# Clinical Relevance of lncRNA and Mitochondrial Targeted Antioxidants as Therapeutic Options in Regulating Oxidative Stress and Mitochondrial Function in Vascular Complications of Diabetes

**DOI:** 10.3390/antiox12040898

**Published:** 2023-04-07

**Authors:** Tarun Pant, Nnamdi Uche, Matea Juric, Zeljko J. Bosnjak

**Affiliations:** 1Department of Medicine, Medical College of Wisconsin, 8701 Watertown Plank Road, Milwaukee, WI 53226, USA; 2Department of Physiology, Medical College of Wisconsin, 8701 Watertown Plank Road, Milwaukee, WI 53226, USA; 3Department of Biophysics, Medical College of Wisconsin, 8701 Watertown Plank Road, Milwaukee, WI 53226, USA

**Keywords:** diabetes, oxidative stress, mitochondria-targeted antioxidants

## Abstract

Metabolic imbalances and persistent hyperglycemia are widely recognized as driving forces for augmented cytosolic and mitochondrial reactive oxygen species (ROS) in diabetes mellitus (DM), fostering the development of vascular complications such as diabetic nephropathy, diabetic cardiomyopathy, diabetic neuropathy, and diabetic retinopathy. Therefore, specific therapeutic approaches capable of modulating oxidative milieu may provide a preventative and/or therapeutic benefit against the development of cardiovascular complications in diabetes patients. Recent studies have demonstrated epigenetic alterations in circulating and tissue-specific long non-coding RNA (lncRNA) signatures in vascular complications of DM regulating mitochondrial function under oxidative stress. Intriguingly, over the past decade mitochondria-targeted antioxidants (MTAs) have emerged as a promising therapeutic option for managing oxidative stress-induced diseases. Here, we review the present status of lncRNA as a diagnostic biomarker and potential regulator of oxidative stress in vascular complications of DM. We also discuss the recent advances in using MTAs in different animal models and clinical trials. We summarize the prospects and challenges for the use of MTAs in treating vascular diseases and their application in translation medicine, which may be beneficial in MTA drug design development, and their application in translational medicine.

## 1. Introduction

Diabetes mellitus (DM) is a disease of global prevalence, and its incidence is steadily increasing [1]. DM can be categorized into four clinically distinct types including type 1 diabetes (T1D), type 2 diabetes (T2D), gestational diabetes (GD), and other specific types, which together affect 425 million people worldwide [2,3]. T1D is a heterogeneous disease characterized by immune-mediated destruction of pancreatic beta cells, resulting in complete insulin deficiency. T1D accounts for 5–10% of the overall cases of diabetes and its progression is majorly driven by both genetics and environmental factors. In contrast, T2D is far more prevalent and constitutes 90–95% of total DM diagnoses. Its development is primarily attributed to insulin resistance, the pathology of which is correlated with lifestyle and environmental changes that have perturbed metabolic pathways, resulting in the elevation of pathophysiological processes associated with T2D susceptibility. Without better interventions, T2D is expected to affect 700 million people in the next several decades [2]. Another class of DM that develops explicitly during second or third trimester of pregnancy, leading to insulin resistance, is GD. Also, in few individuals, specific types of diabetes develop due to other causes, e.g., pre-existing diseases such as pancreatitis and cystic fibrosis, or to the administration of medications like glucocorticoids [3]. DM patients display an increased risk of developing adverse cardiovascular diseases (CVDs) such as valvular heart disease (VHD), ischemic heart disease (IHD), coronary artery disease (CAD), and diabetic cardiomyopathy (DCM), which account for 50.3% of all deaths in diabetic patients [4,5]. Additionally, microvascular (neuropathy, nephropathy, and retinopathy) and macrovascular complications (coronary heart disease, cerebrovascular disease, and peripheral artery disease) are also significant comorbidities affecting 12% of diabetic patients worldwide [5,6]. Considering the consistent increase in diabetes cases and the associated cardiac and vascular complications, it is imperative that we improve our understanding of these underlying pathophysiological mechanisms to develop better diagnostic tools and preventive measures for complications in diabetic patients.

Mitochondria serve as the cell’s powerhouse, as they are the major sites of energy production within the cell. Over the past few years, there has been a significant surge in publications addressing oxidative stress and impairment of mitochondrial homeostasis in diabetes pathogenesis [7,8]. T2D is a chronic heterogeneous disease associated with hyperglycemia, accumulation of advanced glycation end-products (AGEs), elevated fatty acid levels and dyslipidemia, inflammation, elevated production of reactive oxygen species (ROS), and impaired mitochondrial function [9,10,11]. In addition, an immense body of research has focused on ROS-induced damage in endothelial, cardiac, and other cell types, leading to the onset of diabetes-related secondary complications [12,13]. For instance, high levels of glucose, free fatty acids and lipid metabolites such as ceramide in both animal models of diabetes, and in T2D patients, is often deemed to elevate oxidant production, affecting components of the mitochondrial electron transport chain(ETC) to augment ROS production and result in oxidative stress [14,15,16,17,18,19,20]. In fact, this elevated fatty acid level-induced ROS is also found to contribute to DCM development via impairment of mitochondrial function and activation of sensitive proteases that cleave structural proteins, resulting in loss of sarcomere integrity and distortion of the structure of cardiomyocytes (CMs) [21,22,23]. Additionally, dysregulation in peroxisome proliferator-activated receptors (PPARs) expression, responsible for fatty acids (FAs) oxidation in animal models of diabetes, has been reported to enhance mitochondrial ROS generation, mitochondrial respiratory dysfunction, and apoptosis in diabetic complications [24,25,26,27,28]. Moreover, a multitude of studies suggest that oxidative stress may modulate the expression signature of numerous long noncoding RNAs (lncRNAs), contributing to diabetes disease pathogenesis [29,30]. Therefore, it is essential to seek valuable diagnostic biomarkers of oxidative stress and relevant strategies to mitigate mitochondrial oxidative stress to prevent diabetes disease development. To gain insight into the role oxidative stress, mitochondrial dysfunction, and relevance of lncRNAs play in the pathogenesis and diagnosis of DM, as well as applications of conventional and emerging antioxidant modalities in diabetic complications, we used PubMed and Google scholar databases to search literature systematically up to December 2022.

This review presents an overview of the role of oxidative stress and the methods to assess it in diabetes-related complications. Additionally, we discuss the emergence of lncRNA as biomarkers of oxidative stress and their role in modulating redox balance and mitochondrial function in diabetes-related vascular complications. Finally, we discuss the challenges of conventional antioxidants and the potential use of mitochondrial-targeted antioxidants (MTAs) to combat mitochondrial dysfunction and oxidative stress as a therapeutic strategy that may prevent diabetic complications.

## 2. Implication of Oxidative Stress in Diabetes Related Complication

### 2.1. Oxidative Stress and Mitochondria

Oxidative stress refers to the state of redox imbalance between the intracellular production of oxidants, and their neutralization by enzymatic and non-enzymatic antioxidants under physiological or stress conditions [31,32,33]. ROS is a collective term used to define the direct and downstream products formed upon one- or two-electron reduction of molecular oxygen (O_2_) [34]. Many ROS are free radicals, including superoxide radical anion (O_2_^•−^), hydroxyl radical (^•^OH), peroxyl radicals (ROO^•^), alkoxyl radicals (RO^•^), nitrogen dioxide (^•^NO_2_) and carbonate radical anion (CO_3_^•−^). Non-radical ROS include hydrogen peroxide (H_2_O_2_), organic peroxides (ROOH), hypohalous acid (HOCl, HOBr), and peroxynitrite (ONOO^−^) acting as oxidizing, halogenating and/or nitrating species [34,35].

ROS are produced via enzymatic reactions in mitochondria, endoplasmic reticulum (ER), peroxisomes, microsomes, and by NADPH oxidase (NOX) complexes in cellular membranes [36]. Among different intracellular sources of ROS, mitochondria are considered a substantial contributor due to the “electron leak” from the ETC [37,38]. Under aerobic conditions, cells utilize mitochondria as essential organelles to generate energy in the form of adenosine triphosphate (ATP) via a process known as oxidative phosphorylation (OXPHOS). This process involves oxidation of energetic substrates and a transfer of electrons via ETC complexes to reduce O_2_ to water (H_2_O) at complex IV of the ETC. Mitochondrial complexes I-III are not capable of 4-electron reduction of O_2_ to H_2_O, and the transfer of electrons to O_2_ at these complexes results in the formation of O_2_^•−^ [37]. While O_2_^•−^ is likely to be primarily produced by the redox centers of the respiratory chain including complex I (NADH: ubiquinone oxidoreductase) and complex III (ubiquinone: cytochrome c oxidoreductase; cytochrome bc1 complex other potential sources include associated metabolic enzymes like xanthine oxidase (XO), and α-ketoglutarate dehydrogenase [37,38,39,40,41,42,43,44,45]. Of note, the production of O_2_^•−^ within the mitochondria is relatively higher in its resting stage, favored by the high protonmotive force (Δp), NADH/NAD^+^ ratio and reduced coenzyme Q (CoQ) pools [37]. Upon formation, O_2_^•−^ can quickly react with cellular targets such as aconitases or may be converted to other forms of ROS, including the reaction with ^•^NO to form ONOO^−^, or superoxide dismutase-catalyzed or non-enzymatic dismutation to H_2_O_2_. H_2_O_2_ can be further converted to ^•^OH in the presence of transition metal ions, or to an array of oxidizing, halogenating or nitrating species in the presence of peroxidases [39]. 

Prominent non-mitochondrial sources of O_2_^•−^ include the NOX family of transmembrane enzymes NOX isoforms 1–5, and dual oxidases (Duox) 1 and 2 responsible for the generation of O_2_^•−^ and/or H_2_O_2_ via catalyzing the transfer of electrons to O_2_ from NADPH [46,47]. Additionally, accumulating lines of evidence have indicated a positive correlation between protein misfolding-induced ROS generation mediated via protein disulfide isomerase (PDI)-endoplasmic reticulum oxidoreductase (ERO)-1, glutathione (GSH)/glutathione disulfide (GSSG), NADPH oxidase 4 (Nox4), and NADPH-P450 reductase (NPR) in the ER lumen [48,49,50]. Moreover, extracellular sources like ionizing radiation, toxic chemicals (e.g., paraquat), pollutants, and drugs (e.g., Adriamycin, bleomycin) can also stimulate cellular production of ROS [51,52,53].

Under normal physiological conditions, 0.2–2.0% of the total O_2_ consumed by the mitochondria may be reduced to O_2_^•−^/H_2_O_2_ [54,55]. This fraction may further increase under pathological conditions [56,57]. Increased production of ROS may result in modification of various cellular components, resulting in cellular dysfunction and death. Cellular targets of ROS include proteins, lipids, and DNA [58]. Oxidative stress-induced lipid peroxidation forms lipid radicals, peroxyl radical and hydroperoxide, eventually resulting in several low-molecular-weight decomposition products, such as acrolein, malondialdehyde (MDA), and 4-hydroxy-2-nonenal (4HNE). ROS may interact directly with protein amino acids (e.g., oxidation of cysteine thiol group, oxidation and nitration of tyrosine, oxidation of tryptophan), or proteins may be modified by the above-mentioned products of lipid peroxidation. Oxidative post-translational modification of cellular proteins may be involved in redox signaling but may also lead to protein inactivation and/or degradation [59,60]. Interaction of ROS with DNA may result in modification of individual nucleotide bases, single-strand breaks, and cross-linking [61]. 

Under pathological conditions such as glucolipotoxicity, as seen in DM, mitochondria produce excessive ROS that alter mitochondrial dynamics [62]. In the mammalian cell, mitochondrial dynamics comprise mitochondria fission regulated by fission 1 protein (Fis1) and dynamin-related protein 1(Drp1), whereas mitochondrial fusion is modulated by mitofusin2 (Mfn2) and optic atrophy protein 1 (OPA1) [63,64]. Numerous studies have illustrated that oxidative stress developed during DM leads to mitochondrial dysfunction because of dysregulation between fission/fusion and regulatory factors, increasing the production of ROS, which is associated with diabetic complications [65,66]. Furthermore, unbalanced fission/fusion can affect various biological processes, such as apoptosis, leading to different pathological complications. For instance, some studies showed that increased ROS emission mediated mitochondrial fission. At the same time, others indicated that fission was the cause of ROS. Intriguingly, in the past few years, normalization of ROS emission using conventional antioxidants have been reported to mitigate diabetes-related modulating mitochondrial regulatory factors Drp1/Mfn2. Additionally, under oxidative stress, mitochondrial-generated ROS contributes to mtDNA damage in patients with DM characterized by decreased functioning and glucose-stimulated insulin secretion of β cells.

Overall, oxidative stress has been implicated in the development of various diseases, highlighting the importance of improving our understanding of oxidative stress-related mechanisms contributing to the crosstalk between redox imbalance, ROS and mitochondrial dysfunction and their role in vascular diseases in diabetes.

### 2.2. Mediators of Oxidative Stress and Their Role in the Development of Diabetic Complications

Under conditions of cellular homeostasis, ROS has been shown to play an essential role in numerous physiological processes via participating in various signaling networks [67]. However, increased production of ROS has been implicated in the mechanisms of diabetes-induced pathological complications including inflammation, autophagy, fibrosis, necrosis, and apoptosis, impairing the biological function of different organs [9]. Typically, in T2D, a metabolic shift leads to impaired glucose tolerance and lipid metabolism. These may result in hyperlipidemia and consequent accumulation of ectopic lipid metabolites, including diacylglycerol (DAG), long-chain acyl-CoAs (LCACoA), and ceramide [68]. This results in increased free radical production and interferes with insulin signaling in adipocytes and non-adipose tissues like liver and skeletal muscle [69]. For instance, accumulation of DAG in the liver was associated with the activation of protein kinase Cε(PKCε), subsequently reducing the phosphorylation of insulin-stimulated receptor substrate-2(IRS-2) via inhibition of insulin receptor kinase activity, which leads to hepatic insulin resistance [70,71]. Similarly, LCACoA, through stimulation of PKC isoenzymes, disrupts muscle glucose utilization promoting insulin resistance. Additionally, ceramide attenuates insulin signaling and promotes insulin resistance by blocking the transmission via phosphatidylinositol-3kinase (PI3K) and inhibiting the activation of Akt kinase/Protein B(PKB) [72].

The progression of diabetes-related complications is also determined by the intracellular and extracellular production and accumulation of AGEs. Glucose at high concentrations tends to react with free amino groups in the proteins, forming a Schiff base, which fosters the production of AGEs. AGEs impart diverse effects in cells via receptor-dependent- and independent mechanisms thereby contributing to the complications of diabetes by raising intracellular oxidative stress and chronic inflammation [73,74]. For instance, AGE interaction with advance glycation end product-specific receptor (RAGE) on the endothelial cell surface can stimulate NADPH oxidase-mediated ROS production [75]. Additional mechanisms such as AGE-induced ROS generation and the subsequent increased activation of monocyte chemoattractant protein 1 (MCP1), and intracellular adhesion molecule-1 (ICAM-1) accelerated inflammation and were associated with severity of diabetic complications [76,77,78]. Evidence of AGE/RAGE mechanisms exacerbating micro vasculopathy in vivo has also been demonstrated. Diabetic leptin receptor (Lepr^dbdb^) deficient mice displayed increased expression and production of tumor necrosis factor (TNF-α) and ROS, subsequently increasing the transcription factor nuclear factor-κB (NF-κB) [79]. Further, diabetic apolipoprotein (ApoE) deficient mice devoid of RAGE displayed reduced atherosclerosis [80].

Intracellular signal transduction of ectopic lipid metabolites and AGEs increases intracellular ROS. Numerous studies have demonstrated that ROS induced substantial damage to macromolecules, including nucleic acids, lipids, and proteins [61]. For instance, clinical studies have shown elevated levels of oxidized DNA products such as 8-hydroxy-2′-deoxyguanosine (8-OHdG) and 8-oxo-7,8-dihydro-2′-deoxyguanosine(8-oxodG), the products of ROS-induced DNA damage, in the blood and urine specimens of diabetic patients compared to their respective controls [81,82,83,84,85]. Similarly, lipid oxidation products like malondialdehyde (MDA), 4-hydroxynonenal (4HNE), and thiobarbituric acid-reactive substances (TBARS) have been quantified in biological fluids to assess oxidative stress in diabetic patients [86,87,88,89,90]. It should be noted, however, that photometric analyses of TBARS need to be further validated by identification of the products formed before the results can be interpreted as indicative of increased lipid peroxidation. Interestingly, hyperglycemia often alters the activity of critical proteins involved in antioxidant defense, such as superoxide dismutases (SOD), glutathione peroxidases (GPX), and catalase (CAT) which are found to corelate with clinical parameters such as glycated hemoglobin (Hb1Ac) and oxidative stress biomarkers 8-OHdG in diabetic patients [91,92,93,94].

Despite their methodological limitations, both DNA oxidation biomarkers and markers of lipid peroxidation provide experimental support for the involvement of oxidative stress in diabetes and diabetic complications. However, from mechanistic, diagnostic, and therapeutic standpoints, improved assays and oxidative stress biomarkers need to be established and validated for both qualitative and quantitative characterization of the oxidative stress component, with rigorous identification of the products of oxidative modification of biomolecules in collected biological fluids.

### 2.3. Circulating lncRNAs: Emerging Biomarkers of Oxidative Stress in Diabetes Complication

Long non-coding RNAs (lncRNAs) depict the most diverse class of non-coding RNA transcripts, including linear and circular RNAs (circRNAs) > 200 nt in length, with no significant role in translation [95,96]. Current transcriptome profiling and RNA sequencing data from various studies highlight the presence and stability of these lncRNAs in biological fluids, including blood, serum, plasma, and urine, raising the possibility that they might have clinical significance [97,98,99,100]. High-throughput RNA sequencing and profiling data from past studies, including ours, have elucidated the association of aberrant signatures of circulating lncRNA with diabetic complications, establishing their potential value as biomarkers for early prediction of diabetes development and for developing a strategy for its management [101,102]. In this section, we specifically emphasize the circulating lncRNAs implicated with oxidative stress and those which have clinical significance in diabetes-related CVDs (Figure 1).

Recently, several studies investigated changes in the signature of circulating lncRNA and their association with oxidative stress in diabetes-associated complications. For example, patients with diabetic nephropathy (DN) had a significantly higher expression level of circulating lncRNA Metastasis-associated lung adenocarcinoma transcript 1 (MALAT1), also known as NEAT2, which was accompanied by decreased levels of SOD [103]. Also, the expression of lncRNA CASC2 was significantly reduced in the serum of DN patients and human mesangial cells (HMCs) cultured under high glucose (HG) in vitro. Restoration, of the CASC2 expression was found to mitigate HG-induced oxidative stress in vitro via modulating the miR-133b/FOXP1 axis [104]. In addition to DN, impaired circulating lncRNA expression indicates the redox status of diabetes retinopathy (DR) patients. Atef et al. [105] found that the expression of lncRNA HIF1A-AS2 was increased in the serum of proliferative DR patients, and its expression level was found to positively correlate (*p* < 0.05) with predictors of oxidative stress [105]. Recently, Antisense RNA to INK4 locus (ANRIL), also known as CDKN2B-AS1, has been reported to play a significant role in the pathogenesis and development of diabetes-induced complications [106]. Silencing ANRIL expression restored HG-induced oxidative stress in podocytes via upregulating membrane metal endopeptidase (MME) in DKDs [107]. Similarly, lncRNA Brown Fat lncRNA 1(Blnc1) was upregulated in the serum of DN patients compared to matched controls (*n* = 30) and in streptozotocin-induced diabetic mice models and high glucose-induced HK2 cells. Suppression of Blnc1using siRNA in vitro was found to mitigate oxidative stress and inflammation via modulation of NRF2/HO-1 and NF-κB pathways in HK2 cells [108].

Besides the lncRNAs mentioned above, other investigators also found dysregulation in lncRNA expression in diabetic patients with cardiovascular complications. For example, Zhu et al. [109] reported elevated expression of ANRIL in circulating PBMCs of patients with DKDs which showed positive correlation with disease progression [109]. On the other hand, in patients with diabetic lung disease, the serum expression level of smoke and cancer-associated lncRNA 1 (SCAL1) was significantly downregulated. Intriguingly, the expression level of SCAL1 and that of iNOS and NO production depicted an inverse correlation when investigated in HG (30 mM) cultured normal lung cells, which could be attenuated by overexpression of SCAL1 [110]. Furthermore, the level of another lncRNA NEAT1 was found to be high in the plasma of diabetic ischemic stroke patients (DISP) [111]. Again, in T2DM patients, the circulating lncRNAs NKILA, NEAT1, MALAT, and MIAT were highly correlated with the clinical parameters [112].

Altogether, the studies mentioned above indicate that changes in the expression of human lncRNAs can be correlated to oxidative stress and may serve as a diagnostic and prognostic biomarker. However, the inadequacies of the present studies such as (a) insufficient sample size; (b) ambiguity regarding the function of several lncRNAs and the way they modulate oxidative stress, (c) changed expression signature being precisely due to oxidative stress, as opposed to a consequence of diabetes and (d) lncRNA stability in subjects requiring more extended period for sample collection raise few challenges that need to be addressed. Future studies should focus on modifying oxidative stress to quantify lncRNA levels and derive a direct link between them. Furthermore, future studies can be focused on prospectively tracking the changes of these circulating lncRNAs with disease progression, which can then be used to develop a therapeutic intervention to mitigate disease development and progression. (Table 1) summarizes emerging lncRNAs as oxidative-stress biomarkers.

### 2.4. LncRNA as Regulators of Oxidative Stress and Mitochondrial Function in Diabetic Complications

Mitochondria can utilize a dynamic range of substrates for generating ATP through OXPHOS to maintain energy homeostasis [113]. The coding and regulation of the functional proteome machinery of the mitochondria are complicated as its components are derived from both mitochondrial and nuclear genomes [114,115]. The advancement of OMIC and NGS technologies has provided a greater insight into the context of the subcellular localization and transportation of lncRNA transcripts between organelles [116]. Over the past decade, lncRNAs have emerged as a critical regulator of essential cellular processes, including metabolism [117]. Additionally, an emerging line of evidence indicates that mitochondria-associated lncRNA (mtlncRNA) transcripts, including the RNAs derived from mtDNA and nuclear-encoded DNA, work in concert with transcription factors and other epigenetic regulators to modulate oxidative stress and mitochondrial functions [118,119]. Moreover, numerous lncRNA have been cataloged to translocate to the mitochondria and impact its structure, function, and genomic integrity [120]. However, the exact mechanisms via which lncRNA can modulate redox homeostasis and mitochondrial function in diabetes-associated complications remain poorly understood. This section explicitly discusses the pathophysiological relevance of dysregulated lncRNAs in the HG environment and their role in modulating oxidative stress and mitochondrial function (Figure 2).

In recent years several studies showed that dysregulated lncRNAs were involved in regulating redox homeostasis in diabetic complications. For instance, lncRNA Lethe expression was found to be downregulated in RAW264.7 macrophages cultured under high glucose conditions (25 mM) for 24 h. with concomitant increase in ROS production and NOX2 expression. A similar expression of lncRNA Lethe was observed in a mouse model of diabetic wound healing. The overexpression of Lethe in the RAW264.7 cell was found to mitigate hyperglycemia-induced oxidative stress via binding to the p65 subunit of NF-κB and blocking its binding to DNA attenuating NOX2 expression [121]. In another study, Xie and colleagues [122] identified lncRNA Gas5 expression to be downregulated in the HG-stimulated HK2 cells. They also revealed HG upregulated ROS, TNF-α, IL-6, MCP1, and miR-452-5p and decreased the expression levels of SOD in HK2 cells. Meanwhile, Gas5 overexpression in HG-stimulated HK-2 cells transfected with pcDNA-GAS5 was found to revert oxidative stress and pyroptosis, downregulating the expression of miR-452-5p [122]. MALAT-1 is among the most well-studied lncRNA in the nervous system and has numerous neurological functions such as synapse formation and dendrite growth. The impact of MALAT-1 on neurological function was validated in a rat model of diabetes and after gain-and-loss of function studies in HG-cultured brain microvascular endothelial cells (BMEC), demonstrating that inhibiting lncRNA MALAT1 by siRNA lessens apoptosis of microvascular endothelial cells cultured in high glucose via activating the miR-7641/TPR axis [123]. Additionally, LINC01619 lncRNA was downregulated explicitly in podocyte cells, inducing oxidative and ER stress via modulating miR-27a/FOXO1 and subsequently impairing renal function in DN patients [124].

Mechanistic and functional studies on the implication of individual lncRNAs that influence mitochondrial function in diabetic complications are also emerging. Our group was one of the first to show dysregulation of thousands of lncRNA alongside mRNA and its association with DCM development both in diabetic mice and human induced pluripotent stem cell-based disease modeling [125,126]. Interestingly, several dysregulated lncRNAs from our studies were associated with crucial signaling pathways, including TNF-α- and p38 MAPK- that modulate mitochondrial function in DCM [102]. More recently, Zhang et al. [127] described the role of lncRNA LncDACH1 modulating mitochondrial oxidative stress by interacting with sirtuin3 in the DCM heart and HG-stimulated cardiomyocytes. The expression of lncDACH1 was found to be decreased under hyperglycemic conditions, exacerbating mitochondria-derived reactive oxygen species (mtROS) levels, apoptosis and decreasing the activity of manganese superoxide dismutase (Mn-SOD). Meanwhile, the silencing of lncDACH1 attenuated ROS production, mitochondrial dysfunction, cell apoptosis, and increased the activity of Mn-SOD in cardiomyocytes treated with HG [127]. Following the studies above, nuclear encoded lncMALAT1 and lncNEAT1 were reported to translocate to the mitochondria of HG cultured human retinal endothelial cells (HRECs), affecting mitochondrial structure and function and leading to oxidative stress. Modulating the expression of lncMALAT1 and lncNEAT1 by their respective siRNA restored mitochondrial homeostasis by mitigating oxidative stress [128]. 

In summary, deciphering the epigenetic modulations and mechanism of action of lncRNAs implicated in oxidative stress-induced mitochondrial dysfunction might pave the path for lncRNA-based therapeutics to prevent diabetic complications. Different dysregulated lncRNAs and their role in modulating mechanisms associated with oxidative stress and mitochondrial function are summarized in (Table 2).

### 2.5. Conventional Antioxidants and Their Application in Diabetic Complications: Success and Limitations

Cells have complex antioxidant systems consisting of both enzymatic and nonenzymatic processes that work harmoniously to protect the organism against damage caused by free radicals. Unfortunately, these antioxidant defenses may be downregulated in people with diabetes, as suggested by the reports of reduced plasma/serum status of total antioxidants, scavenging activity of free radicals, certain antioxidant enzymes and levels of specific antioxidants, including vitamin E and ascorbic acid [129]. The resultant augmentation of oxidative stress, combined with the association between diabetes susceptibility risk and antioxidant intake or serum levels, would suggest that mitigating oxidative stress via treatment with antioxidants may be an effective strategy for preventing or managing diabetic complications. Indeed, many animal studies involving treatment with various antioxidants have demonstrated reversals in diabetes-induced changes of oxidative stress indicators. However, a cautionary note in interpreting the results of these studies would be to consider the extensive assortment of the experimental model of diabetes, duration of the studies, and type of antioxidant used in addition to the markers that were evaluated for oxidative stress. Moreover, the results of many studies may need to adequately reflect the actual serum antioxidant status as they are focused on only a limited number of individual antioxidants [130]. There are also different categories of antioxidants: those that prevent the initiation of radical chain reaction and those that interrupt the oxidative process altogether by interacting with chain-propagating radicals to form stable products [131]. Further complicating matters is that their respective activities or impact can be measured regarding their binding capacities or serum concentrations. Accordingly, caution may be warranted when interpreting results depending on the type of antioxidant studied and the assay performed. While studies with experimental models have shown favorable effects of antioxidants reducing diabetic complications, evidence for using classic antioxidant regimes from clinical trials has demonstrated variable and conflicting results. For example, in a prospective cohort study, intake of vitamin C was found to be associated with reduced development of diabetes [132] and potentially being effective against complications faced by diabetics [133]. However, these findings are discounted by the reports of unsuccessful clinical trials, even trials with observed increases in plasma vitamin C [134,135,136]. N-acetylcysteine (NAC) is an L-cysteine prodrug whose antioxidant properties are due to its ability to replenish levels of intracellular glutathione (GSH). It’s the only antioxidant drug to have clinical utility in improving a renal condition whose pathologic mechanism may involve oxidative stress. A pilot study demonstrated that NAC could not reduce serum biomarkers of oxidant stress in patients with diabetic proteinuria [134] even though preclinical studies have shown cardiovascular benefits or reduced progression of diabetes attributed to NAC-mediated inhibition of ROS production [137,138,139]. Intake of the fat-soluble vitamin E, known to be deficient in diabetes, was reported to be significantly associated with reduced risk of type 2 diabetes [140]. While clinical trials demonstrated favorable effects of its supplementation on biomarkers of oxidative stress [141,142] other studies have failed to show beneficial effects on diabetic complications and clinical events [143,144,145]. A constant among these conventional antioxidants is that they work by directly scavenging the already present oxidants in a stoichiometric manner. The relative quantities or ratio of antioxidants to oxidants highlights a fundamental shortcoming of antioxidant therapy. Additionally, introduction of an antioxidant into the already highly oxidative environment of a diabetic patient may inadvertently promote the pro-oxidant characteristic of the antioxidant [146].

Diabetic patients have been reported to display abnormally high ROS production and experience reduced expression of antioxidant enzymes [9]. Another approach to ameliorating oxidative stress would involve the induction of the cell’s own endogenous antioxidant response system. The Nrf2/Keap1/ARE pathway is a fundamental protective response pathway to oxidative stress. Induction of this pathway results in the promoted expression of several cytoprotective and antioxidant response genes which function to counter reactive molecules [147]. Activation of this pathway and resultant increased expression of many antioxidant response enzymes would presumably afford more promising results than supplementation of antioxidants that only scavenge ROS in a stoichiometric manner. Indeed, preclinical studies and clinical trials have suggested that activation of this pathway can reverse or delay the oxidative stress-induced diabetic dysfunction. Clinical trials with Nrf2 activators having shown initial promise were not, however, without their shortcomings. A phase III trial (BEACON) with a potent Nrf2 activator, bardoxolone methyl, in TIIDM patients with stage IV CKD had to be terminated early due to severe adverse events including death. While potential off-target effects of bardoxolone methyl should not be disregarded, another consideration is that the cell’s own endogenous antioxidant response system may not be sufficient to protect the cell against reactive molecules. Moreover, it has been suggested that Nrf2 activation be done in a timely and highly specific manner with consideration for cell types that would be affected and induction of target genes [148]. A cautionary note is that while generally resulting in cytoprotective effects, the temporal expression of the many different Nrf2-regulated genes may confer differential responses that may or may not benefit the cell. Additionally, antioxidant enzymes and proteins generally rely on an optimal amount of cellular reducing equivalents (i.e., NADH, NADPH, low molecular weight thiols including GSH) for their ability to function effectively. The supply of reducing equivalents defines the ultimate antioxidant capacity of a tissue [149]. Such reducing equivalents may be mostly consumed during oxidative stress, thus compromising the cell’s capacity for antioxidant activities. Perhaps a more comprehensive approach would involve ensuring an adequate supply of reducing equivalents to guarantee an effective antioxidant response and prevent the system from shifting too far into the oxidative stress side of the redox spectrum. That is to say that maintaining an appropriate amount of the components necessary for an efficacious antioxidant enzyme activity may provide the cell a better capacity to prevent oxidative stress. Most of the reactive molecules would be scavenged and Nrf2 activation would not be warranted. In this context, the lack of Nrf2 activation would be a consequence of having an optimal cellular redox status as opposed to the interpretation that the lack is harmful.

## 3. Mitochondrial-Targeted Antioxidants for Diabetic Complications

Mitochondria are recognized as critical sites of localized injury in several chronic pathologies, leading to the development of organelle-directed therapeutics. Many pharmacological strategies have been developed over the past two decades to mitigate mtROS [150,151]. One of the approaches employed is to specifically target a range of active compounds, including antioxidants, to the mitochondria, via their conjugation with a delocalized lipophilic cation (DLC) such as triphenylphosphonium (TPP) (Figure 3).

Intriguingly, the DLC is hydrophobic and can freely pass through the phospholipid bilayer of the plasma membrane and other organelles without the requirement for a specific uptake mechanism. Additionally, the transmembrane potential (Δ*Ψ*m) across the plasma membrane (−30 to −60 mV) and the mitochondrial inner membrane (−150 to −170 mV) drives the movement of lipophilic TPP conjugated with compounds from the extracellular space into the cytoplasmic compartment and then specifically to the mitochondrial matrix [152,153]. This has led to the development of a wide range of bioactive molecules and probes that can be delivered to mitochondria in vivo following oral, intravenous, intraperitoneal, and subcutaneous administration [154]. Studies showed low antioxidant activity and increased free radicals in patients with diabetes even before diagnosis. Therefore, mitochondrial-targeted antioxidants (MTAs) have been intensively investigated using animal models, cell culture-based studies, and clinical trials to restore redox homeostasis in diabetic complications. Representative studies on the TPP-linked MTAs (Figure 4) observed to play a role in protection against different complication induced by diabetes are summarized in (Table 3).

### 3.1. MitoQ

Mitoquinone (MitoQ) is among the most substantially utilized antioxidant molecules targeted to the mitochondria. It is the structural analog of the endogenous antioxidant Coenzyme Q10 (CoQ10), also known as ubiquinone, attached to TPP by a hydrophobic 10-carbon alkyl chain [155]. These properties enable MitoQ to cross the plasma membrane and accumulate in the mitochondrial matrix [156]. It also allows the ubiquinone component to access the active site of complex II where it is converted to the active antioxidant, ubiquinol, to reduce mtROS. Preclinical findings from recent studies have demonstrated that targeted mitochondrial therapy using the antioxidant MitoQ has proven to be beneficial for treating diabetes-related cardiovascular complications. The therapeutic efficacy of MitoQ has been investigated in mice models of diabetic kidney diseases (DKDs), where intraperitoneal administration of MitoQ reverted tubular injury by ameliorating mtROS and mitochondrial fragmentation via enhancing mitophagy mediated by Nrf2 and PTEN induced kinase 1 PINK1 [157]. Ward et al. [158] also suggested that MitoQ can mitigate mitochondrial dysfunction, conferring equivalent renoprotection in DKDs as conventional first-line therapies like ramipril [158]. Another interesting proof of action that treatment with MitoQ can be protective in diabetes was documented by assessing its effect on pancreatic β cells cultured under normoglycemic (NG, 11.1 mM glucose), hyperglycemic (HG, 25 mM glucose) and lipidic (palmitic acid (PA), 0.5 mM) conditions. MitoQ was found to prevent augmentation of ROS production, O_2_ consumption, protein levels of ER stress markers (GRP78 and P-eIF2α), and the proinflammatory nuclear transcription factor NFκB-p65 under HG condition via modulating mitochondrial function, which in turn facilitates endoplasmic reticulum stress improving β cell function [159]. In the neuroprotective assessment of MitoQ in vivo, Fink et al. [160] reported dietary supplementation of MitoQ (0.93 g/kg diet) for 12 weeks improved peripheral neuropathy in in vivo models of diet-induced obesity and T2D [160]. Yang et al. [161] also reported that MitoQ treatment exerts a protective effect by reducing HG-induced ROS, cytoskeletal damage, and apoptosis in brain microvascular endothelial cells (BMECs) via activating the Nrf2/HO-1 pathway [161]. Additionally, Ji et al. [162] documented that MitoQ was able to ameliorate post-myocardial-ischemia reperfusion (MIR) injury in T2D by mitophagy upregulation via PINK1/Parkin pathway in rat models of T2D [162]. Moreover, the antioxidant and anti-inflammatory action of MitoQ was recently reported in leukocytes whereby its administration reduced ROS production, leukocyte-endothelium interactions, and TNFα through the action of NFκB, preventing CVDs in T2D patients [163].

### 3.2. SKQ1

10-(6′-plastoquinone)decyltriphenylphosphonium (SKQ1s) is another efficient antioxidant, structurally similar to MitoQ but, with replacement of ubiquinone with plastoquinone that has been successfully targeted to the mitochondria. SkQ1 is one of the most prevalent utilized SKQs, having a TPP moiety covalently conjugated with plastoquinone [164]. In previous studies, Voronkova et al. [165,166] evaluated the ability of SkQ1 to restore hyperglycemia-induced oxidative stress in diabetic rat models via attenuating mtROS, and decreasing the burden on the antioxidant machinery such as catalase, superoxide dismutase, glutathione, and NAPDH-generating enzymes [165,166]. In support of this finding, Agarkov et al. [167] also demonstrated that free radical fluctuation was significantly alleviated in the heart and blood serum of rats with streptozotocin-induced hyperglycemia upon administration of SkQ1 [167]. Recently, studies have reported the beneficial effects of antioxidant pre-therapy using SkQ1 in diabetes-related complications. For example, Demyanenko et al. [168] observed that diabetic C57BL/KsJ-db^−^/db^−^ mice orally supplemented with SkQ1 (250 nmol/kg of body weight) for eight months showed increased wound healing via suppressing lipid peroxidation and immune cells (neutrophils and macrophages) infiltration [168]. In addition, SkQ1 was also effective in improving the blood glucose level in diabetic rats via genetic/epigenetic modifications in the expression of mRNA/microRNA associated with oxidative stress and metabolisms [169].

### 3.3. Mito-Tempo

Mito-Tempo is another cell-permeable antioxidant with a robust radical scavenging property targeted explicitly to mitochondria, via piperidine nitroxide (TEMPO) conjugation with the TPP moiety. Mito-Tempo is a mitochondria targeted SOD mimetic that can catalytically convert O_2_^•−^ to O_2_ and H_2_O_2_ and has been shown to alleviate mitochondrial induced-oxidative stress in diabetes-associated cardiovascular complications [170]. Regarding cardiomyopathy, treatment with Mito-Tempo, diminished high glucose-induced oxidative stress and apoptosis, which abrogated myocardial dysfunction in diabetic mice via modulating the ERK1/2 pathway [171]. In a T1D rat model associated with vascular dysfunction, a reduction in vascular tone, hypertension, and ROS production were prevented by treatment with Mito-Tempo via modulation of the GLP-1/CREB/adiponectin axis [172]. Mito-Tempo also significantly improved endothelial function by reducing mitochondrial ROS in the arterioles of T2D patients [173]. Additionally, Mito-Tempo administration restored insulin resistance-induced impairment of mitochondrial complex II activity in the visceral adipose tissue of patients with severe obesity [174].
antioxidants-12-00898-t003_Table 3Table 3Mitochondrial targeted antioxidants in different diabetic complications.Mitochondria Targeted AntioxidantsExperimental ModelsDosageEffect/MechanismLimitationReferences MitoQMice model of diabetes kidney disease 5 mg/kg × bodyweight; intraperitoneal administrationReverts tubular injury by ameliorating mtROS and mitochondrial fragmentation via activating NRF2 and PINK1.
[157]Mice model of diabetic kidney disease 0.6 mg/kg × bodyweight; intragastric gavageMitigates mitochondrial dysfunction and conferred renal protection. Contrasting effect on AER and ACR as response to therapies.[158]Pancreatic β cell line INS-1E model of HG0.5 µmol/L in culture medium Protects the β cell via preventing ROS production and decreasing endoplasmic reticulum stress and NFκB-p65 activation under high glucose.
[159]Mice model of peripheral neuropathy 0.93 g/kg × bodyweight; diet administration Increases motor and sensory nerve conduction velocity cornea sensitivity and thermal nociception improving peripheral neuropathy. 
[160]Brain microvascular endothelial cells (BMECs) model of HG50 µmol/L in culture medium Attenuates mitochondrial ROS production, cytoskeletal damage and apoptosis in BMEC via activating Nrf2/HO-1 pathway. 
[161]Rat model of myocardial ischemia reperfusion injury2.8 mg/kg × bodyweight; tail vein administrationConfers cardio protection by promoting mitophagy via modulating PINK1/Parkin pathway.
[162]Leukocytes of T2D patients 0.5 µmol/L in culture medium Decreases ROS production, leukocyte endothelium interaction, TNFα in the leukocytes of T2D patients via regulating NF-κB pathway.Small cohort size and lack of control group with similar BMI as T2D patients[163]SKQ1Rat model of protamine sulfate induced hyperglycemia1250 nmol/kg × bodyweight;IntraperitonealadministrationDecreases ROS production, free radical oxidation and restored total antioxidant activity and mitigates hyperglycemic stress.
[165,166]Rat model of streptozotocin induced hyperglycemia1250 nmol/kg × bodyweight;IntraperitonealadministrationDecreases free radical production and oxidation restoring the antioxidant activity of catalase and SOD to the direction of control to mitigate hyperglycemic stress.
[167]Mice model of diabetic dermal wound healing 250 nmol/kg × bodyweight;Oral administration Increases mitochondrial biogenesis, normalizes inflammation enhancing wound healing 
[168]Mito-TempoMice model of diabetic cardiomyopathy0.7 mg/kg × bodyweight;IntraperitonealadministrationDecreases mitochondrial reactive oxygen species generation decreasing apoptosis and myocardial hypertrophy via regulation ERK1/2 pathway. 
[171]Rat model of diabetic induced vascular constriction 20 mg/kg × bodyweight;IntraperitonealadministrationAttenuates abnormal vascular tone and hypertension via modulation of GLP-1/CREB/adiponectin pathway.
[172]Arterioles and mononuclear cells of T2D patients1 mmol/L in culture medium Improves endothelial function and reduces mitochondrial superoxide levels.Small study size, medication effects influencing mitochondrial function of mononuclear cells were not excluded [173]Visceral adipose tissue of T2D patients 10 µmol/L in culture medium Restores the activity of mitochondrial complex II and insulin sensitivity.Small sample size, inherent variability in complex II activity between subjects and markers of mitochondrial functions were not measured [174]Abbreviation: T2D:Type 2 diabetes; MTA: Mitochondrial targeted antioxidants; PINK1: PTEN induced kinase 1; NOX2: NADPH oxidase 2; ERK1/2: Extracellular signal-regulated protein kinase 1/2; GLP-1: Glucagon-like peptide 1; CREB: cAMP response element binding protein; GRP78: Glucose related protein 78; NFκB: Nuclear factor kappa B; Nrf2: Nuclear factor erythroid 2–related factor 2; PTEN: Phosphatase and tensin homolog; HO-1: Heme oxygenase 1; TNFα: Tumor necrotic factor α; GPx: Glutathione peroxidases; GSTs: Glutathione S transferase; GSRs: Glutathione disulfide reductase; GLP-1: Glucagon-like peptide 1; CREB: Cyclic AMP response element binding protein.


### 3.4. Mito-PBN

Mito-PBN has emerged as a promising MTA preventing oxidative stress by scavenging mitochondria-generated free radicals. Recently, Wu et al. generated and intraperitoneally administered liposome encapsulated nano-Mito-PBN particles to assess its therapeutic effect in restoring redox balance in the liver of animal models of diabetes. The results of the present study indicated that liposomal encapsulated Mito-PBN resulted in a relatively higher accumulation in hepatocytes, efficiently scavenging mitochondrial O_2_^•−^ and H_2_O_2,_ resulting in a decreased ratio of NADH: NAD+, improved mitochondrial oxidative energy coupling, and ATP synthesis, which alleviated ROS-induced mitochondrial dysfunction [175].

### 3.5. Szeto-Schiller (SS) Peptides

SS-peptides are another class of cell-permeable MTAs that have been shown to play a protective role against mitochondrial-generated ROS in several pathological conditions, including diabetes. Administration of mitochondria targeted SS-31 peptides in vitro and in vivo has been found to restore mitochondrial function in numerous diabetic complications, which has been reviewed in the past [176] and subsequently lies outside the scope of this review article. However, future studies in animal models of diabetes are warranted to better assess treatment efficacy before designing clinical trials. 

In conclusion, based on the data generated from numerous preclinical studies, MTAs can be considered a class of compounds with the potential to alleviate oxidative stress and prevent diabetes-related cardiovascular complications. However, clinical studies are warranted to assess the reproducibility of the preclinical findings to choose the most suitable MTAs to alleviate mitochondria-related oxidative stress in diabetic patients.

## 4. Challenges and Prospects of Mitochondrial Targets Antioxidants

Extensive evidence from the past have linked oxidative stress-induced mitochondrial dysfunction to the progression of various cardiovascular complication in diabetic patients [177,178]. Therefore, developing an efficient therapeutic strategy to alleviate mitochondrial oxidative stress is imperative. Antioxidants such as vitamin C, vitamin E (α-tocopherol) and NAC have been utilized in studies with the aim of ameliorating mitochondrial ROS, and while they have been observed to decrease O_2_^•−^ reaction-products, these antioxidants were plagued with shortcomings such as low bioavailability and subpar ROS reduction. To overcome these challenges, researchers sought to target ROS at its main source, the mitochondria, with the implementation of targeted antioxidants (MTAs) able to accumulate within the mitochondria at 10–100-fold higher capacities compared to non-targeted antioxidants. Modern strategies targeting antioxidants specifically to mitochondria have offered an excellent alternative to mitigate oxidative stress and restore redox balance, improving mitochondrial function in different diseases, including diabetes. Additionally, high selectivity, fewer side effects, ease of production, efficient pharmacokinetics, and absorption make them an ideal tool to be tested in clinical settings. This targeted approach has aided common pharmacological criteria desired for drug development including targeted action, increased potency and clinically relevant drug doses. However, despite extensive preliminary data, substantial challenges like variation in the mitochondrial membrane potential which may vary in different diabetic complications and is confounded by non-specific or lower mitochondrial accumulation, clinically relevant dosage, and nonspecific interaction of TPP cations with mitochondriotropic agents, needs to be taken into consideration before assessing their effects in clinical trials. Additionally, it has been a point of discussion that the ability of the MTAs to restore the redox balance is due to its effects as a radical scavenger or because of its ability to modulate the enzyme activity of other antioxidants. Therefore, we hypothesize that modification of the structure of MTAs to modulate their radical scavenging activity while retaining similar pharmacokinetic properties may help decipher their underlying therapeutic mechanisms to mitigate oxidative stress. 

## 5. Conclusions and Perspectives

Oxidative stress is now widely recognized as a crucial factor contributing to the development and progression of diabetes-related complications. Therefore, it is relevant to identify oxidative stress biomarkers and the methods to measure them to develop a comprehensive understanding of the link between oxidative stress biomarkers and disease development and progression in diabetic patients. In recent years, the expression level of lncRNA has shown a strong correlation with oxidative stress indicating that they might modulate redox status in diabetic complications. Furthermore, there is emerging scientific evidence depicting that MTAs can mitigate mitochondrial oxidative stress caused by hyperglycemia in diabetes, providing promising oxidative stress-targeting therapeutic strategies. However, most studies to date investigating the effect of MTAs on oxidative stress have been conducted using animal models or cell cultures rather than diabetic patients.

In the future, it is imperative that further research focuses on designing clinical studies assessing the safe use of MTAs and their beneficial effects in diabetic patients. Furthermore, combining MTAs in conjunction with lncRNA-targeting strategies shows promise as a potential therapeutic strategy for diabetes and diabetic cardio- and vascular complications.

## Figures and Tables

**Figure 1 antioxidants-12-00898-f001:**
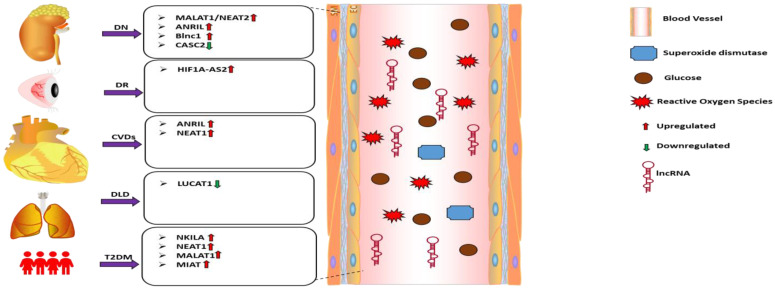
Roles and functions of the listed lncRNAs in the regulation of oxidative stress in diabetic complications. Arrows indicate the direction of changes in the expression or function. DN: Diabetic nephropathy; DR: Diabetic retinopathy; CVDs: Cardiovascular diseases; DLD: Diabetes lung disease; T2DM: Type 2 diabetes mellitus; lncRNA: Long non-coding RNA; MALAT1: Metastasis-associated lung adenocarcinoma transcript 1; ANRIL: Antisense non-coding RNA in the INK4 locus; Blnc1:Brown Fat lncRNA 1; CASC2: Cancer susceptibility candidate 2; HF1A-AS2: Hypoxia inducible factor 1 alpha-antisense RNA 2; NEAT1: Nuclear paraspeckle assembly transcript 1; LUCAT1: Lung cancer associated transcript 1; NKILA: NF-κB-interacting long noncoding RNA; MIAT: Myocardial infarction-associated transcript.

**Figure 2 antioxidants-12-00898-f002:**
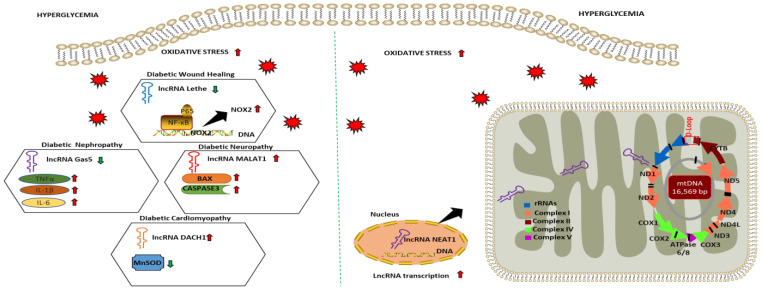
Dysregulated lncRNA and their role in regulating oxidative stress and mitochondrial function in diabetic complications. Arrows indicate the direction of changes in the expression or function. lncRNA: Long non-coding RNA; Gas5: Growth arrest specific transcript 5; DACH1: Dachshund homolog 1; NEAT1: Nuclear paraspeckle assembly transcript 1; NOX2: NAPDH Oxidase 2; NF-κB: Nuclear factor-kappa B; TNFα: Tumor necrotic factor alpha; IL1-β: Interleukin-1 beta; IL6: Interleukin 6; BAX: Bcl-2-associated X protein; MnSOD: Manganese superoxide dismutase; ND: NADH dehydrogenase; COX1: Cytochrome c oxidase 1 rRNA: Ribosomal RNA.

**Figure 3 antioxidants-12-00898-f003:**
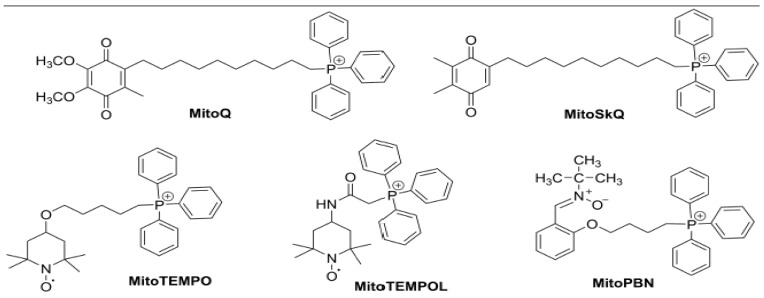
Chemical structures of TPP-linked mitochondrial-targeted antioxidants MitoQ, SkQ1, Mito-TEMO, MITO-TEMPOL and Mito-PBN.

**Figure 4 antioxidants-12-00898-f004:**
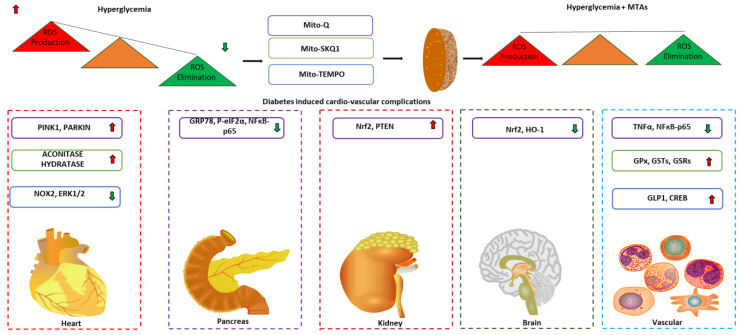
Potential application of mitochondrial targeted antioxidants in diabetic complications. Arrows indicate the direction of changes in the expression or function. MTA: Mitochondrial targeted antioxidants; PINK1: PTEN induced kinase 1; NOX2: NADPH oxidase 2; ERK1/2: Extracellular signal-regulated protein kinase 1/2; GRP78: Glucose related protein 78; NFκB: Nuclear factor kappa B; Nrf2: Nuclear factor erythroid 2–related factor 2; PTEN: Phosphatase and tensin homolog; HO-1: Heme oxygenase 1; TNFα: Tumor necrotic factor α; GPx: Glutathione peroxidases; GSTs: Glutathione S transferase; GSRs: Glutathione disulfide reductase; GLP-1: Glucagon-like peptide 1; CREB: Cyclic AMP response element binding protein.

**Table 1 antioxidants-12-00898-t001:** lncRNAs implicated in the pathogenesis of oxidative stress in different diabetic complications.

Diabetic Complications	lncRNA	Study Design/Sample Size	Biological Fluid or Cells	lncRNA Expression Level	Interacting Factors/Relevant Pathways	Oxidative Stress Markers	References
Diabetic Nephropathy	MALAT1	Diabetic Nephropathy patients (*n* = 47)	Serum	Upregulated		SOD	[103]
Diabetic Nephropathy	CASC2	Diabetic Nephropathy patients (*n* = 27)	Serum and HG-treated mesangial cells	Downregulated	miR-133b/FOXP1	SOD MDA	[104]
Diabetic Retinopathy	HF1A-AS2	Diabetic Retinopathy patients (*n* = 60)	Serum	Upregulated	MAPK/VEGF	ONOO^−^ NO MDA	[105]
Diabetic Kidney disease	ANRIL	Diabetic Kidney disease patients (*n* = 22)	Serum and HG-treated podocytes	Upregulated			[107]
Diabetic Nephropathy	Blnc1	Diabetic Nephropathy patients (*n* = 30)	Serum and HG-treated HK2 cells	Upregulated	NRF2/HO-1 NF-κB		[108]
Diabetic Kidney disease	ANRIL	Diabetic Kidney disease patients (*n* = 21)	PBMC	Upregulated			[109]
Diabetic Lung Disease	SCAL1	Diabetic Lung disease patients (*n* = 56)	Serum and HG-treated lung cells	Downregulated		NO iNOS	[110]
Diabetic Ischemic Stroke	NEAT1	Diabetic Ischemic stroke patients (*n* = 22)	plasma	Upregulated	miR-124		[111]

Abbreviation: lncRNA: Long non-coding RNA; MALAT1: Metastasis-associated lung adenocarcinoma transcript 1; CASC2: Cancer susceptibility candidate 2; HF1A-AS2: Hypoxia inducible factor 1 alpha-antisense RNA 2; Blnc1: Brown Fat lncRNA 1; ANRIL: Antisense non-coding RNA in the INK4 locus; SCAL1: Smoke and cancer associated lncRNA1; NEAT1: Nuclear paraspeckle assembly transcript 1; HG: High glucose; PBMC: Peripheral mononuclear cells; FoxP1: Forehead box protein 1; Nrf2: Nuclear factor erythroid 2–related factor 2; HO1: Heme oxygenase 1; NF-κB: Nuclear factor-kappa B; SOD: Superoxide dismutase; MDA: Malonaldehyde; NO: Nitric oxide; iNOS: Inducible nitric oxide synthase.

**Table 2 antioxidants-12-00898-t002:** lncRNAs and the mechanisms associated with oxidative stress and mitochondrial function in diabetic complications.

Diabetic Complications	lncRNA	Experimental Model	Expression Level	Function	Oxidative Stress/Cell Viability Markers	References
Diabetic Wound Healing	Lethe	HG-treated RAW264.7	Downregulated	Increases NOX 2 expression and ROS production via modulating NF-κB signaling	Intracellular ROS	[121]
Diabetic Nephropathy	Gas5	HG-treated HK2	Downregulated	Increases oxidative stress and pyroptosis via modulating miR-452-5p	MDA SOD	[122]
Diabetic Neuropathy	MALAT1	HG-treated BMEC	Upregulated	Promotes cellular apoptosis via upregulating miR-7641/TPR expression	BAX CASPASE3	[123]
Diabetic Nephropathy	LIN01619	HG-treated podocytes cells	Downregulated	Augments ER stress via modulating miR-27a/FOXO1	Intracellular ROS	[124]
Diabetic Cardiomyopathy	DACH1	HG-treated cardiomyocytes	Upregulated	Increases mitochondrial-derived ROS, mitochondrial dysfunction and cellular apoptosis via increasing SIRT3 degradation	MnSOD	[127]
Diabetic Retinopathy	MALAT1 NEAT1	HG-treated HREC	Downregulated	Dysregulates mitochondrial homeostasis by damaging mitochondrial structure and genome integrity	mtROS	[128]

Abbreviation: lncRNA: Long non-coding RNA; HG: High glucose; Gas5: Growth arrest specific transcript 5; MALAT1: Metastasis-associated lung adenocarcinoma transcript 1; DACH1: Dachshund homolog 1; RNA 2; NEAT1: Nuclear paraspeckle assembly transcript 1; HK2: Human kidney 2; BMEC: Bone marrow microvascular endothelial cells; HREC: Human retinal endothelial cells; FoxO1: Forehead box protein O1; NF-κB: Nuclear factor-kappa B, NOX2: NAPDH oxidase 2; TPR: Translocated promoter region, Nuclear basket protein; SOD: Superoxide dismutase; MnSOD: Manganese superoxide dismutase; MDA: Malonaldehyde; BAX: Bcl-2-associated X protein; mtROS: Mitochondrial reactive oxygen species.

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
