# Peer review of "Clinical Relevance of lncRNA and Mitochondrial Targeted Antioxidants as Therapeutic Options in Regulating Oxidative Stress and Mitochondrial Function in Vascular Complications of Diabetes"

_antioxidants, 2023, doi:10.3390/antiox12040898_

Round 1

Reviewer 1 Report

The review entitled “Clinical relevance of lncRNA and mitochondrial targeted anti-oxidants as therapeutic options in regulating oxidative stress and mitochondrial function in vascular complications of diabetes” by Pant et al., represent a good review for the pathogenesis of DM and the role of oxidative stress and mitochondrial dysfunction in the development of diabetic complications.

It highlights the role of lncRNA in pathogenesis and diagnosis as well as the antioxidant modalities to prevent and treat diabetic complications.

Abstract

Line 15, 20 vascular diseases better replaced by vascular complications.

Line 20 DMs better to use DM.

Introduction

Line 32 types of DM please include other types like gestational DM and write one sentence about Type 1DM pathogenesis and complications.

Line 38

 {Developing adverse cardiovascular diseases (CVDs) such as valvular heart disease (VHD), ischemic heart disease (IHD), coro-39 nary artery disease (CAD)}.

Please mention macrovascular complications as you described the microvascular ones in the next paragraph.

Line 47 {diagnosis and prevention in diabetic patients}

Prevention of what? development of prediabetes into overt diabetes or prevention of diabetic complications? Please specify.

Line 52 AGE, better to use AGEs.

Line 63 and 64 CMs, FAs, please write the full names before using the abbreviation.

Methodology of retrieving the review would better be introduced like; which databases were used for literature search and the range of years.

Line 67, (recent research), the references are 2019, 2021, so please use other word instead of (recent).

Line 94; NOX abbreviation should be replaced before not after in the text.

Lines 119-122, the author would like to link the mitochondrial dysfunction to endoplasmic stress so please rephrase this paragraph in better wording.

Under the section of oxidative stress and mitochondria

The authors would better add paragraph describing the effects of oxidative stress on mitochondrial morphology (fission and fusion events), mitochondrial quality control gene expression and mitochondrial DNA damage. Only brief mention of DNA modification was mentioned in lines 138 and 139.

Line 151, please mention type 2 diabetes.

Line 152

a metabolic shift leads to insulin resistance, impaired glucose tolerance, and increased lipid metabolism.

This sentence is not clear please describe the underlying mechanisms of insulin resistance and the word increased lipid metabolism is incorrect better to use impaired.

Please mention the changes in lipid metabolism in DM specifically T2DM.

More details about hyperglycemia induced AGEs, and how they bind to their receptors (RAGE) and initiate inflammatory signaling like NF-kB that accelerates diabetic macro and micro-complications and augments oxidative stress.

Lines from 151-160 need to be rephrased and incorporate the above-mentioned part.

Line 212, Atef et al., please add reference 90 after the author’s name.

Line 215 recently is bold please change, the reference 91 might be incorrect because it is 2008 and you mentioned recently so, please adjust.

Line 197 figure 1 has no legend.

Line 246 table 1 has no caption.

Also, the reference 91 in table 1 is irrelevant to diabetic kidney diseases, I think you mean reference no. 93 which is more relevant.

Figure 2 there is no legend and the figure resolution made it hard to review.

Resolution of figures and tables need to be improved. All other figures in the manuscript have no legends please add description to each figure and abbreviations used.

Line 278 should be NF-kB, please correct.

Lines 444- 465 please remove bold fonts and revise grammar. Please do so in the whole text.

Line 401

3. Mitochondrial Targeted Antioxidants: Therapeutic Implication for Attenuating mitochondrial induced oxidative stress in diabetic complications

This title is lengthy please make it shorter.

Figures 3 and 4 lack legends.

Lines 444-465 correct bold fonts, revise grammar. please do so in the whole manuscript.

In the whole manuscript list the reference number after the author’s name. e.g., line 460 Ji et al. ……..,  please add reference number.

Lines 565-568 please rephrase.

References

As mentioned earlier the authors should specify the years of literatures used in the current study.

Ref. 34, 35, 42   and others could be better replaced by recent articles if possible.

Author Response

Please find the itemized response in the PDF

Reviewer 2 Report

This is an interesting review article providing a perspective on the oxidative stress biomarkers and mitochondria-targeted antioxidants in the vascular complication of DM. The title is useful in the field of antioxidants.

It would be very nice if the author discusses more about why mitochondria-targeted antioxidants are superior to the conventional antioxidants in reducing oxidative damage.

Author Response

Please find the itemized response in the PDF document.

Reviewer 3 Report

This manuscript reviews the involvement of Lnc RNAs in diabetic complications and potential therapeutic use of mitochondrial-targeted antioxidants in diabetes

Content

Sections 2.1 and 2.2 are not novel or even updated. It is better to refer the reader to more detailed and updated reviews.

Limitations of the studies which suggested the involvement of Lnc RNAs in diabetic complications should be discussed.

Is there any relation between Lnc RNA and MTAs

Figures

Figures are not clear in terms of organization and resolution, no caption provided, no reference to the figures in the text, no explanation for abbreviation.

Tables

A table should be prepared to compare different MTAs, their doses, experimental models that were used, limitations of the studies in which they were used.

Table 1 Add a column to clarify if studies were done on humans or experimental animals

A separate table should be added for Lnc RNAs which were confirmed to target mitochondrial function

Abbreviation

A list of abbreviations should be added before the introduction.

In instances, an abbreviation is used without explanation e.g. Line 94, in other instances an abbreviation is explained twice e.g. DCM in lines 40 and 299

Abbreviations should be explained in the footnotes of each figure or table

English

There are numerous English mistakes, so a careful review is required e.g. line 84  omit “the”

Author Response

(The authors gave the same response as above.)

Round 2

Reviewer 3 Report

The list of abbreviations is not complete e.g. T1D is present in the text but not included in the list. Also, the abbreviations are not arranged alphabetically in the list.

There are still English language mistakes in the manuscript.

Author Response

Please find attached the itemized response. 
